Phenotypic characterization of patients with deletions in the 3’-flanking SHOX region

Kant Sarina G. s.g.kant@lumc.nl 1
Broekman Sander J. 2
de Wit Caroline C. 2
Bos Marloes 1
Scheltinga Sitha A. 2
Bakker Egbert 2
Oostdijk Wilma 3
van der Kamp Hetty J. 3
van Zwet Erik W. 4
van der Hout Annemieke H. 5
Wit Jan M. 3
Losekoot Monique 2
1 CHCG-Department of Clinical Genetics, Leiden University Medical Center , Leiden , The Netherlands
2 CHCG-Laboratory for Diagnostic Genome Analysis, Leiden University Medical Center , Leiden , The Netherlands
3 Department of Pediatrics, Leiden University Medical Center , Leiden , The Netherlands
4 Department of Medical Statistics, Leiden University Medical Center , Leiden , The Netherlands
5 Department of Genetics, University Medical Center Groningen, University of Groningen , Groningen , The Netherlands
Meyre David
Electronic publication date: 2013 Feb 19
Publication date: 2013
Volume: 1
Electronic Location ID: e35
Received 2012 Nov 21; Accepted 2013 Jan 18
Copyright: © 2013 Kant et al.
Copyright year: 2013
Copyright holder: Kant et al.
License: This is an open access article distributed under the terms of the Creative Commons Attribution License, which permits unrestricted use, distribution, and reproduction in any medium, provided the original author and source are credited.
License URL: https://creativecommons.org/licenses/by/3.0/

Keywords: Downstream deletion, Phenotype, SHOX, Short stature, Disproportion, Madelung

Funding: No grants or fellowships funded the research or the writing of this article.

==============================
Context. Leri–Weill dyschondrosteosis is a clinically variable skeletal dysplasia, caused by SHOX deletion or mutations, or a deletion of enhancer sequences in the 3’-flanking region. Recently, a 47.5 kb recurrent PAR1 deletion downstream of SHOX was reported, but its frequency and clinical importance are still unknown.

Objective. This study aims to compare the clinical features of different sizes of deletions in the 3’-flanking SHOX region in order to determine the relevance of the regulatory sequences in this region.

Design. We collected DNA from 28 families with deletions in the 3’-PAR1 region. Clinical data were available from 23 index patients and 21 relatives.

Results. In 9 families (20 individuals) a large deletion ( ∼ 200–900 kb) was found and in 19 families (35 individuals) a small deletion was demonstrated, equal to the recently described 47.5 kb PAR1 deletion. Median height SDS, sitting height/height ratio SDS and the presence of Madelung deformity in patients with the 47.5 kb deletion were not significantly different from patients with larger deletions. The index patients had a median height SDS which was slightly lower than in their affected family members (p = 0.08). No significant differences were observed between male and female patients.

Conclusions. The phenotype of patients with deletions in the 3’-PAR1 region is remarkably variable. Height, sitting height/height ratio and the presence of Madelung deformity were not significantly different between patients with the 47.5 kb recurrent PAR1 deletion and those with larger deletions, suggesting that this enhancer plays an important role in SHOX expression.

Introduction

Leri–Weill dyschondrosteosis (LWD) is caused by haploinsufficiency of SHOX in 60–70% of cases (intragenic deletions, duplications or mutations), and in 15–31% of patients without an intragenic mutation or deletion a putative enhancer sequence in the 3’-flanking region has been demonstrated (Benito-Sanz et al., 2005; Chen et al., 2009). SHOX is located on the short arm of the X-chromosome in the pseudoautosomal (PAR) 1 region, thereby escaping X-inactivation, leading to a pseudoautosomal pattern of inheritance for LWD (Belin et al., 1998; Shears et al., 1998). So far, 7 enhancer sequences, 3 upstream and 4 downstream of SHOX, have been described (Fukami et al., 2006; Sabherwal et al., 2007; Durand et al., 2010; Kenyon et al., 2011; Benito-Sanz et al., 2012a; Benito-Sanz et al., 2012b). Deletion of one or more of the downstream enhancer sequences results in SHOX haploinsufficiency and LWD. Benito-Sanz et al. (2005) as well as Chen et al. (2009) described a cohort of patients with different deletions in the downstream enhancer region, where various deletion sizes all resulted in LWD or idiopathic short stature. A deletion in the enhancer region upstream of SHOX has only recently been reported in one female patient and her father with idiopathic short stature (Benito-Sanz et al., 2012a), where two of the three upstream enhancer sequences were deleted.

LWD caused by a heterozygous mutation or deletion of SHOX is characterized by short stature, mesomelia, and Madelung deformity. Penetrance of the phenotype appears to be incomplete within families (Rappold et al., 2007; Binder, 2011) and Madelung deformity and short stature tend to be more common and severe in females than in males (Binder, 2011). Mean height of index cases has been reported as −2.2 SDS (Binder, 2011; Rosilio et al., 2012) and height of affected mothers and fathers as −2.1 SDS and −1.6 SDS, respectively (Rosilio et al., 2012).

There is limited information about the phenotype of patients with a deletion in the enhancer region downstream of SHOX. However, from the available reports the phenotypic characteristics do not seem to differ much from those of patients with SHOX deletions or mutations (Benito-Sanz et al., 2005; Salmon-Musial et al., 2011; Rosilio et al., 2012).

Recently, a relatively small deletion of PAR1 was found by Caliebe et al. (2012) in a father and son, of whom the son had short stature (−2.0 SDS), while the father did not (−0.44 SDS). Comparable deletions were already demonstrated by Chen et al. (2009), but their exact size was not determined in detail. Subsequently this deletion was described in 30 unrelated families with LWD or ISS by Benito-Sanz et al. (2012b) equivalent to 15.3% of probands with LWD and 1.9% of probands with idiopathic short stature, uncovering a novel downstream enhancer. The clinical phenotype of these patients showed a wide variability with height SDS ranging from −0.14 to −4.68.

We aimed at investigating the frequency of the various deletions in the 3’-flanking SHOX region and the possible phenotypic differences between patients with different sizes of deletions in this region.

Methods

Clinical characteristics

In 49 DNA samples sent to the Laboratory for Diagnostic Genome Analysis of the Department of Clinical Genetics (Leiden University Medical Center, Leiden) and in six of the DNA samples sent to the Section Genome Diagnostics of the Department of Genetics (University Medical Center Groningen, Groningen), for the determination of SHOX defects between 2005 and 2011, we found a deletion in the downstream enhancer region of SHOX in 29 probands, and 26 of their relatives (total of 28 families). In the same period in the Laboratory for Diagnostic Genome Analysis of the Department of Clinical Genetics (Leiden University Medical Center, Leiden) we detected 43 deletions of the complete gene, 22 mutations (missense, nonsense, splice site or frameshift) and 5 duplications of SHOX in a total of 1045 patients.

With consent of the Medical Ethical Committee of the LUMC, clinical data were collected and anonymized for 44 of the 55 patients (23 index patients and 21 affected relatives from 23 families). Two of the remaining 11 patients were excluded from the analysis of clinical features, because they suffered from an additional disorder, which would probably influence their growth (a skeletal dysplasia of unknown type (inconsistent with LWD), and a deletion 22q11, respectively). For 9 patients no additional clinical data could be obtained.

Demographic and clinical data collected were sex, age, height standard deviation score (SDS), sitting height to height ratio SDS and presence of Madelung deformity. Height SDS and sitting height/height SDS were calculated based on Dutch nation-wide references (Fredriks et al., 2005). We chose not to collect all data needed to calculate the Rappold score (Rappold et al., 2007), because of the limitations of this scoring system in clinical practice: arm span, and length and bowing of the forearm are seldom measured by clinicians, appearance of muscular build is not very often spontaneously mentioned in clinical records, and the use of body mass index gives a higher score for obese patients. Instead, we chose to collect data about the classical clinical triad of Leri–Weill dyschondrosteosis: height, sitting height to height ratio (SH/H) and presence of Madelung deformity.

Molecular analysis

DNA isolation from peripheral blood and sequence analysis of the SHOX coding region including intron–exon boundaries was performed using standard procedures (primer sequences available upon request). Deletions and duplications were detected using the MRC-Holland MLPA kit (Salsa P018-D1) according to the manufacturer’s instructions.

Statistical analysis

The t-test and Fisher exact test were used to test for an association of genotype (deletion including PAR1 probes L05099-L05101 or larger deletions) with height SDS, SH/H SDS, and the presence of a Madelung deformity. Differences between index patients and their relatives were determined, as well as the influence of the gender of the patient.

Results

Of the 28 families, 19 had the recurrent 47543 bp (47.5 kb) deletion encompassing PAR1 probes L05099-L05101 described by Benito-Sanz et al. (2012b), containing a SHOX enhancer sequence (ECR1/CNE7). The deletion was found in 19 index cases and 16 of their relatives. Nine families (10 index patients and 10 relatives) had a larger deletion varying from a deletion that includes PAR1 probes L05096-L05104 to a deletion including PAR1 probes L05098-L05103 (Fig. 1).

Figure 1 Schematic representation of the SHOX locus and the surrounding enhancer regions.

Schematic representation of the SHOX locus and location of the 3’-flanking PAR1 probes of the P018-D1 MLPA kit (MRC Holland). Extent of the deletions (blue bars) and the number of families (n) is indicated with the number of affected individuals between brackets. The SHOX gene is indicated with a purple box. The PAR1 probes are indicated with an * and the four downstream enhancers (ECN4, ECN5, ECR1/CNE7 and CNE9) are indicated with green or red (the enhancer within the 47.5 deletion) triangles. The deletion described by Sabherwal includes PAR1 probes L05096-l L05101.

In the 44 individuals (23 families) of whom clinical information was available, mean height SDS was −1.9 (range −4.0 to 1.2) and SH/H SDS 1.5 (range −1.6 to 3.0). Madelung deformity was present in 19 of 33 patients. Mean height SDS and SH/H SDS in patients with the 47.5 kb deletions (n = 25) compared to patients with larger deletions (n = 19) are shown in Fig. 2. Patients with the 47.5 kb deletion were on average 0.5 SD shorter than patients with larger deletions, but the difference did not reach statistical significance (p = 0.20). A comparison of the same clinical characteristics between the index patients and their affected relatives, and between male and female patients, is also shown in Fig. 2. No significant differences were observed in the different subgroups of patients, as illustrated in Figs. 2 and 3. Presence or absence of a Madelung deformity in the subgroups of patients mentioned above is summarized in Table 1. No significant differences were observed in any of the subgroups.

Figure 2 Height (SDS) range in different subgroups of patients.

Height (SDS) range in different subgroups of patients with a deletion in the downstream enhancer region of SHOX. del 47.5 kb = deletion that includes PAR1 probes L05099-L05101 in the downstream enhancer of SHOX; other del = larger deletion in the downstream enhancer of SHOX; ∗ = mean value; n = number of tested individuals; p = p-value measured by t-test.

Figure 3 Sitting height to height ratio (SDS) range in different subgroups of patients.

Sitting height to height ratio (SDS) range in different subgroups of patients with a deletion in the downstream enhancer region of SHOX. del 47.5 kb = deletion that includes PAR1 probes L05099-L05101 in the downstream enhancer of SHOX; other del = larger deletion in the downstream enhancer of SHOX; ∗ = mean value; n = number of tested individuals; p = p-value measured by t-test.

Table 1 Madelung deformity in different subgroups of patients with a deletion in the downstream enhancer region of SHOX.

Three comparisons are made in subgroups of patients. del 47.5 kb = deletion that includes PAR1 probes L05099-L05101 in the downstream enhancer of SHOX; other del = larger deletion in the downstream enhancer of SHOX.

	Madelung present	Madelung absent	p-value (Fisher exact test)	
Patients with del 47.5 kb (n = 17)	11	6	0.49	
Patients with other del (n = 16)	8	8		
Index patients (n = 19)	10	9	0.72	
Relatives (n = 14)	9	5		
Male patients (n = 12)	7	5	0.72	
Female patients (n = 21)	14	7		

Discussion

We show that the recently reported recurrent 47543 bp (47.5 kb) deletion (Benito-Sanz et al., 2012b) could also be demonstrated in 19 out of 28 Dutch families (68%) with a downstream deletion and has a similar clinical phenotype as larger deletions in the downstream region of SHOX. Height SDS, SH/H SDS and the presence of Madelung deformity all showed a large variation, regardless of deletion size, gender of the patient or whether the patient was an index patient or relative.

We have now shown that these deletions comprise an important part of our patient cohort. The lack of previous reports on this deletion could be due to doubt about the pathogenity of the deletion, because it is not overlapping the formerly described critical region in the enhancer region (Benito-Sanz et al., 2005). However, recently the same research group convincingly reported this deletion to include a previously uncharacterized SHOX enhancer (ECR1/CNE7), proving the deletion to be pathogenic (Benito-Sanz et al., 2012b).

The average height SDS in our index cases (−2.2 SDS) is similar to mean height SDS in cohorts of patients with SHOX deletions and mutations (Binder, 2011). In the first report on patients with a deletion in the enhancer region Benito-Sanz et al. (2005) demonstrated 12 families with short stature and/or Madelung deformity with variable severity, with a height SDS ranging from −4.6 SD to 0.85 SD. Rosilio et al. (2012) recently reported a mean height in their patients with a deletion encompassing SHOX of −2.3 SDS, while in the same study a mean height of −2.2 SDS was seen in patients with any deletion in the downstream region. In a smaller sample of patients with a deletion downstream of SHOX in the study of Salmon-Musial et al. (2011) a mean height SDS of −1.4 SDS was demonstrated. The results of these reports are in accordance with our observations, although in the two last mentioned reports it is suggested that the phenotype in patients with a deletion in the downstream enhancer region seems to be milder than in patients with a SHOX mutation or deletion. The six patients described by Salmon-Musial et al. (2011) with anomalies downstream of SHOX had a less severe short stature (height SDS −1.4) than those with other gene anomalies (height SDS −2.5; n = 16). Due to the small size of their sample they were not able to draw firm conclusions. Rosilio et al. (2012) reported a higher percentage of idiopathic short stature (instead of LWD) in patients with a deletion in the downstream enhancer region interval compared to patients with a SHOX mutation or deletion, suggesting a milder phenotype in patients with a downstream deletion. However, classification of patient groups in this study shows some shortcomings, as also acknowledged by the authors, and comparisons must be taken with caution. Furthermore in contrast to the findings of Rosilio et al. (2012); Chen et al. (2009) reported a higher percentage of downstream deletions in LWD patients compared to patients with idiopathic short stature. Therefore, genotype–phenotype correlations do not seem obvious.

Comparing all index patients with their relatives shows that mean height SDS is slightly higher in relatives, although not significantly different (p = 0.08), probably due to referral bias. Differences between male and female patients were not observed in our study, contrary to what is known about patients with SHOX deletions or mutations where females tend to present with a more severe phenotype than males (Binder, 2011). However, further confirmation is needed on a larger patient cohort.

The large variability of the phenotype confirms the findings in previous reports on patients with a deletion in the enhancer region 3’ of SHOX (Benito-Sanz et al., 2005; Chen et al., 2009; Kant et al., 2011; Salmon-Musial et al., 2011; Rosilio et al., 2012). Not all index patients had short stature, which was also noticed in the study by Salmon-Musial et al. (2011), and has been observed in previous studies (Binder et al., 2004; Rappold et al., 2007). Looking in detail at this group of eight patients with a height at last examination above −2 SDS, reason for a diagnostic SHOX analysis was the presence of Madelung deformity in five of them, and disproportion in combination with familial disproportionate short stature in one patient. Another patient had a recent height measurement of −1.4 SDS, but former height measurements were always around −2 SD, while several family members had disproportionate short stature. The last patient had a height in the normal range, but below her target height range. So, short stature is not always the leading clinical sign to request SHOX analysis, and referring clinicians do take into account other clinical characteristics of the patient and their relatives. The variability of the clinical phenotype is not limited to deletions of the enhancer region. Also for LWD caused by SHOX mutations or deletions, the phenotype has been described as highly variable (Binder, 2011). Thus, penetrance of the phenotype appears to be incomplete, even within families.

In conclusion, the phenotype in patients with a deletion in the downstream enhancer region of SHOX seems comparable to that of patients with a SHOX mutation or deletion, and the variability of the phenotype in the whole group of patients with a deletion in the enhancer region is remarkable. The recurrent 47.5 kb deletion downstream of SHOX leads to a phenotype comparable to that of larger deletions in the same region. Our data support that the 47.5 kb deletion is a pathogenic deletion, and therefore it seems plausible that an important enhancer region is located within the deletion interval.

The authors thank all referring clinicians for providing clinical information about their patients, and V Janmaat for his contribution to the data collection.

Additional Information and Declarations

Competing Interests

Author Contributions

JM Wit is an Academic Editor for PeerJ.

Sarina G. Kant conceived and designed the experiments, performed the experiments, analyzed the data, wrote the paper.

Sander J. Broekman, Caroline C. de Wit, Sitha A. Scheltinga, Wilma Oostdijk and Hetty J. van der Kamp performed the experiments.

Marloes Bos performed the experiments, analyzed the data.

Egbert Bakker and Annemieke H. van der Hout analyzed the data.

Erik W. van Zwet conceived and designed the experiments, analyzed the data.

Jan M. Wit and Monique Losekoot conceived and designed the experiments, analyzed the data, wrote the paper.

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
