# Peer review of "Phenotypic characterization of patients with deletions in the 3’-flanking SHOX region"

_PeerJ, doi:10.7717/peerj.35_

## Round 0.1 · original submission · Major Revisions

Dear authors,
Thank you to submit this interesting paper to PeerJ. Please revise the article taking into account the inspired suggestions of the reviewers. The paper can be simplified and significantly shortened.
Best regards,
David Meyre

·

Basic reporting

See General Comments

Experimental design

See General Comments

Validity of the findings

See General Comments

Additional comments

The present study by Sarina Kant et al analysed deletion sizes within the pseudoautosomal region in 28 families with Leri-Weill dyschondrosteosis (LWD), with the prior information that the SHOX gene, which normally causes this disorder, was not affected. They found in roughly two-thirds of their analysed cases a small recurrent deletion of 47kb, thus confirming previous results by Benito-Sanz et al, 2012. Genotype- phenotype comparison then revealed that patients with this small deletion were not significantly different than those with a large deletion.

Specific comments:

1. In the Introduction the authors state that 15% of LWD can be explained by enhancer deletions outside the SHOX gene (this reference relates to the Benito-Sanz et al 2012 paper, not the 2005 paper). Chen et al, 2009, however, found that enhancer deletions are present in 45% of patients with LWD.
2. Page 3: Rosilio et al 2012 reported a higher percentage of downstream deletions in ISS patients compared to LWD patients, while Chen et al , 2009 reported a higher percentage of downstream deletions in LWD patients compared to ISS patients. Therefore , genotype-phenotype correlations seem not obvious.
3. Page 4 :delete „small“ mutations
4. Discussion: Small deletions have been also previously been detected in 4/5 patients with ISS and SHOX downstream deletions and 1/2 patients with SHOX downstream deletions (Figure 2; Chen et al, 2009), but their exact size was not determined in detail
5. Page 9, line 170, … with a deletion of the downstream enhancer interval
6. Figure 1 could be improved by also indicating the enhancer intervals as published by Sabherwal et al, 2007

Reviewer 2 ·

Basic reporting

This article describes the clinical characterization of 23 index patients and 21 relatives with PAR1 deletions located downstream of SHOX, comparing those with a large downstream deletion and those with a recently described 47.5kb deletion. This is an interesting article with valuable clinical evaluation but: 1) the manuscript is too long with a large degree of repetition between the introduction and the discussion, and some discussion points are also made in the results section; 2) the article is not fluid to read. Thus, I would suggest a reduction in length and improvement in the writing style would greatly aid the fluidity of the article.

Major points
Table 1: this table is not clearly described and the numbers do not add up, for example, 12 male patients, 12 have Madelung deformity and 5 do not. These table row headings should be clearly written and the data revised.

Line 45: They state “Benito-Sanz et al reported 12 families with short stature and/or Madelung deformity with variable severity, without further details about height SDS. This isn´t strictly correct as the height SDS´s are listed in Table 1 and range from -4.6 to 0.8 SDS.

Lines 50 and 169: Caution must also be undertaken with the data reported by Rosilio et al, 2012. In this study, they included 140 children from a previous study (Huber et al, 2006), 56 dyschondrosteosis and 84 ISS.
a) The LWD/ISS classification was determined by clinical examination and did not include a routine wrist radiography to determine if they had signs of the Madelung deformity. They only undertook this after the post-genetic analysis and found that 6 of the 18 ISS patients with SHOX defects actually had a Madelung deformity and thus should have been classified as LWD.
b) They identified a total of 10 large deletions using microsatellite marker and SNP analysis. The microsatellite markers were not informative at the 5´ deletion extensions which were within SHOX thus these patients may actually have deletions that include SHOX and thus the clinical evaluation of the SHOX deletions v downstream deletions may have been different.
c) They also identified a common downstream 10.5 kb deletion in their patients, which was subsequently characterized and reported to be a non-pathogenic CNV (Benito-Sanz, Am J Med Genet 2011).
Thus, the conclusions stated in this manuscript from this article have to be taken with caution.

Line 112: Of the 28 families analyzed, 19 index cases had the recurrent 47.5kb downstream deletion whilst 10 index cases (from 9 families) had a larger deletion. Clinical data was available for statistical assessment for a total of 44 individuals from 23 families but the deletion type that these clinically evaluated individuals/families had was not described (line 112). This data is only visible in Figure 2 (25 with the 47.5 kb deletion and 19 with other downstream deletions), but for clarity it should also be included in the text.

Minor points
Line 19: Haploinsuficiency is of a protein not of a gene thus it should be written: “haploinsufficiency of SHOX (not italics)” rather than “haploinsufficiency of SHOX (italics)” throughout the manuscript.

Line 19-20: They state that 60-70% of cases are due to intragenic deletions or mutations but they need to include intragenic SHOX duplications.

Line 31-33: They describe a deletion in the enhancer region upstream of SHOX – they should state that it was in a patient with ISS not LWD.

Line 123 and 176: I don´t think it is now remarkable that not all index patients had short stature. This has been reported numerous times and is now an accepted fact in the clinical characteristics of LWD. These sentences need to be modified.

Line 123: This paragraph is more discussion than results.

Results: When describing which MLPA probes are deleted they should write “PAR1 probes L05096-L05104 inclusive”.

Line 115 they should use “47.5 kb deletion” rather than deletions.

Line 116, the grammar needs to be corrected from “0.5 SD less short” to “0.5 SDS shorter”.

Line 140: “However, we have now shown that these deletions (i.e. 47.5 kb deletion) comprise the major part of our patient cohort”. In the manuscript they state that during the same time period they detected: 43 SHOX deletions, 22 point mutations/small insertions or deletions in SHOX, 5 SHOX duplications and 29 downstream deletions (of which 19 had this 47.5 kb deletion). Thus, this statement is confusing because more SHOX deletions have been identified than this downstream 47.5 kb deletion

Line 156: The grammar needs to be corrected: “mean height SDS is slightly taller in relatives” to “slightly higher”

Line 188-190: the data presented in this manuscript does indeed support the pathogenicity of the 47.5kb deletion but it does not confirm that an important enhancer region is located within the deletion interval as no functional data is presented. The final summary needs to be reworded.

There are many citations that are incorrectly cited or older citations are not included. For example:
Line 24: Rao et al, 1997 detected pseudoautosomal pattern of inheritance for ISS and Turner syndrome but not LWD. They then cite Kant et al, 2011 and Evers et al, 2011 but there are earlier articles that should be cited.
Lines 123 and 176: they cite that not all index patients had short stature, which was also noticed in the study by Salmon-Musial et al 2011. This has been observed in previous studies.

Fig 1: I would remove the critical deletion interval reported by Benito-Sanz et al, because this plays no role in this work. What is important is that the four downstream enhancers are indicated in the figure so that you can see how many and which enhancers are deleted in the three detected deletions.

Experimental design

The design is well set out and evaluated correctly.

Validity of the findings

No comments

---

## Round 0.2 · accepted · Accept

The authors carefully took into account the reviewer's comments and significantly improved the manuscript. The current version of the article is suitable for publication in PeerJ.